# Fusion of Low-Entanglement Excitations in 2D Toric Code

Jing-Yu Zhao[1], Xie Chen,[2]

**1** Institute for Advanced Study, Tsinghua University, Beijing 100084, China
**2** Institute for Quantum Information and Matter and Department of Physics,
California Institute of Technology, Pasadena, CA, 91125, USA

September 13, 2024

## Abstract

On top of a $D$-dimensional gapped bulk state, Low Entanglement Excitations (LEE) on $d(< D)$-dimensional sub-manifolds can have extensive energy but preserves the entanglement area law of the ground state. Due to their multi-dimensional nature, the LEEs embody a higher-category structure in quantum systems. They are the ground state of a modified Hamiltonian and hence capture the notions of 'defects' of generalized symmetries. In previous works, we studied the low-entanglement excitations in a trivial phase as well as those in invertible phases. We find that LEEs in these phases have the same structure as lower-dimensional gapped phases and their defects within. In this paper, we study the LEEs inside non-invertible topological phases. We focus on the simple example of $\mathbb{Z}_2$ toric code and discuss how the fusion result of 1d LEEs with 0d morphisms can depend on both the choice of fusion circuit and the ordering of the fused defects.

# 1 Introduction

The notion of Low Entanglement Excitations (LEE) was introduced in Ref. [1,2] to describe entanglement area law preserving $d$-dimensional excitations on top of a $D$-dimensional gapped ground state, $d < D$. In quantum condensed matter systems, we are usually interested in low energy excitations because they contribute to linear response of the system under external perturbation. The LEE, however, have extensive energy if they live on a submanifold of dimension $d > 1$. While LEEs with $d > 1$ do not contribute to linear response of the system, they can be interesting for the following reasons:

1. The LEEs are ground states of a modified Hamiltonian with terms on the $d$-dimensional sub-manifold different from the original Hamiltonian. Therefore, they correspond to the notion of 'defects' that plays a central role in the definition of generalized symmetries [3–14]. Calling them 'excitations' rather than 'defects' puts emphasis on the dynamical properties of these objects.

2. Higher-dimensional topoloigcal phases with $D \geq 3$ contain elementary fractional excitation like flux loops excitations with $d = 1$ or membrane excitations with $d = 2$. Nonelementary LEEs of the same dimension can generically appear alongside such elementary excitations, and in general it is impossible to completely separate them and identify the 'pure' elementary excitations without the accompanying non-elementary LEEs. An understanding of the structure of all LEEs is hence necessary for the proper description of the bulk excitations in higher-dimensional topological phases.

3. Given their low entanglement, the LEEs are potentially 'condensable' such that their condensation can still have low entanglement and potentially realize a different phase. Therefore, we expect LEEs to play an important role in the description of zero-temperature quantum phase transitions.

In Ref. [1], we studied the LEEs of a trivial phase where the ground state can be a product state. The LEEs on top of a product state are not entangled with each other or with the bulk, and hence correspond to lower-dimensional gapped phases. In particular, we focused on one-dimensional gapped phases and the zero-dimensional domain walls (morphisms in math language) within and studied their fusion using explicit lattice models and quantum circuits. The fusion of one-dimensional phases is achieved with one-dimensional finite depth quantum circuits while the fusion of the domain walls is achieved with zero-dimensional local unitary transformations. The fusion pattern revealed in Ref. [1] is part of the 2-category structure [7, 11, 12, 14–17] formed by one-dimensional gapped phases.

In Ref. [2], we extended the discussion to invertible phaes, such as the symmetry-protected topological phases and the $p + ip$ superconductor. Using the idea of symmetric Quantum Cellular Automata, a 'pumping' process through higher dimensional bulk, as well as the Topological Holography formalism, we showed that the $d$-dimensional LEEs in invertible phases have the same structure as those in trivial phases and hence form the same higher-category structure as $d$-dimensional gapped phases.

In this paper, we study the LEEs in non-invertible topological phases. We focus on simple cases like the $\mathbb{Z}_2$ Toric Code in $D = 2$ and $D = 3$ dimensions and study LEEs of dimension $d \leq 1$. Our discussion is going to be based on the same quantum circuit principles as stated above:

1. Two 1d LEEs are equivalence if they are connected by a 1d finite depth circuit.

2. 1d LEEs generated (from the ground state) by a 1d finite depth circuit are trivial; 1d LEEs that can only be generated (from the ground state) with a 2d circuit or a 1d sequential circuit are nontrivial.

3. Fusion of two 1d LEEs is achieved with a 1d finite depth circuit.

4. Two 0d domain walls on top of a 1d LEE are equivalent if they are connected by a 0d unitary transformation.

5. 0d domain walls that cannot be generated with a 0d unitary transformation are nontrivial.

6. Fusion of two 0d domain walls on top of a 1d LEE is achieved with a 0d unitary.

7. Fusion of two 1d LEEs with nontrivial 0d domain walls on either (or both) is achieved with the same 1d finite depth circuit as the no domain wall case except at the local region near the domain walls. Extra 0d unitary transformations can be applied near the domain walls.

Following these rules, we can get a complete list of 1d LEEs and their 0d domain walls. We study their fusion by constructing explicit 1d circuits or 0d unitary transformations that realize the fusion. We observe several interesting features in the process:

1. Different morphisms between two 1d LEEs can often be distinguished by the half-braiding or full-braiding of bulk anyons around the morphism.

2. The 1d circuit used to fuse 1d LEEs (without morphisms) is not unique. Different circuits can lead to different fusion results when the 1d LEEs to be fused has nontrivial morphisms on top.

3. The fusion of the Cheshire string with possible domain walls is not symmetric under the exchange of the two strings. This is true not only in 2D Toric Code, but in 3D Toric Code as well.

The paper is organized as follows: In Section 2, we review the 1d LEEs in the 2D Toric Code, including their classification and fusion rules through quantum circuits. We also explore the 0d domain walls and endpoints associated with these 1d LEEs, along with their corresponding fusion rules. Section 3 addresses the fusion of 1d LEEs with 0d domain walls or endpoints. Making use of examples such as Cheshire strings and duality strings, we demostrate how different 1d fusion circuits can result in distinct fused 0d morphisms. In Section 4, we provide a disscusion of the results by looking at the 3D Toric Code. Finally, in Section 5, we summarize our conclusions and suggest potential directions for future research.

## 2 LEEs in 2D Toric Code

The classification of 1d LEEs within a 2D Toric Code is already discussed in Ref. [3, 18]. It was further shown in [19] that all of these nontrivial 1d LEEs can be generated by sequential quantum circuits [20]. In this section, we provide a brief review of the 1d LEEs in the 2D Toric Code and their fusion rules in the language of quantum circuits. We also examine the domain walls and boundaries (morphisms) of these 1d LEEs, as well as the fusion of these domain walls and boundaries using local unitary transformations.

Consider the Toric Code on a two-dimensional square lattice

$$H = -\sum_v A_v - \sum_p B_p$$
$$= -\sum_v \prod_{v \in e} X_e - \sum_p \prod_{e \in p} Z_e \, ,$$

(1)

where the summation runs over the vertices $v$ and the plaquettes $p$ of the square lattice. It is well known that the Hamiltonian (1) hosts a $\mathbb{Z}_2$ topological order, with its anyon excitations being the 0d LEEs on top of a trivial 1d LEE: the trivial excitation 1, the charge excitation $e$ that violates the vertex term $A_v$, the magnetic flux $m$ that violates the plaquette term $B_p$, and the fermion excitation $f$ that violates both $A_v$ and $B_p$. The topological nature of these 0d anyons can be seen from their stability under local unitary transformations. For instance, the $-1$ braiding sign between $e$ and $m$ remains invariant under any local unitary evolution around the 0d LEEs. Therefore, we can define two 0d local excitations as equivalent if they can be connected by some 0d local unitary transformation. This is exactly the notion of "superselection" sector defined by Kitaev [21].

## 2.1 1d LEEs and their fusion rules

Making use of similar ideas, we can extend the classification of anyons to 1d LEEs according to their behavior under 1d finite depth circuits. As already mentioned in the introduction, we define two 1d LEEs as belonging to the same equivalent class if they can be connected by a 1d finite depth local unitary quantum circuit. Historically, 1d LEEs are constructed by altering the Hamiltonian along a string [22], and they are regarded as defects in the topologically ordered states. It was shown in Ref. [19] that a linear depth sequential unitary circuits [20] also enable the generation of nontrivial 1d LEEs.

In this paper, we mainly focus on deriving the fusion rules of 0d and 1d LEEs using local unitaries and 1d circuits. Fig. 1 illustrates the Hamiltonian terms stabilizing different 1d LEEs in 2D Toric Code. For instance, consider the Cheshire string $rr$ where the charge $e$ is condensed along it. The condensation can be achieved by first removing the vertex terms $A_v$ along a string, and then adding a polarization term $-Z$ on every bond of the string, as shown in Fig. 1(b). Every $-Z$ term creates a pair of charge $e$ at the two vertices of the bond. The fact that $-Z$ is enforced as a Hamiltonian term means that the charge $e$ is condensed along the dashed line in the ground state.

The six inequivalent types of 1d LEEs in the 2D Toric Code are, as originally outlined in [18]:

1. The trivial string 1, which is the same as the bulk.

2. The Cheshire string $rr$, which can be achieved by condensing the charge $e$.

3. The flux version of the Cheshire string, denoted as $ss$, where the flux $m$ is condensed.

4. Different anyons can condense on different sides of a string. Consequently, there is also a 1d LEE denoted as $rs$, which means that the charge $e$ is condensed on the upper side of the string and the flux $m$ is condensed on the lower side of the string.

5. Similarly there is $sr$, where the flux $m$ is condensed on the upper side of the string and the charge $e$ is condensed on the lower side of the string.

6. Finally, here is a duality string $dual$, through which the charge $e$ and flux $m$ are exchanged.

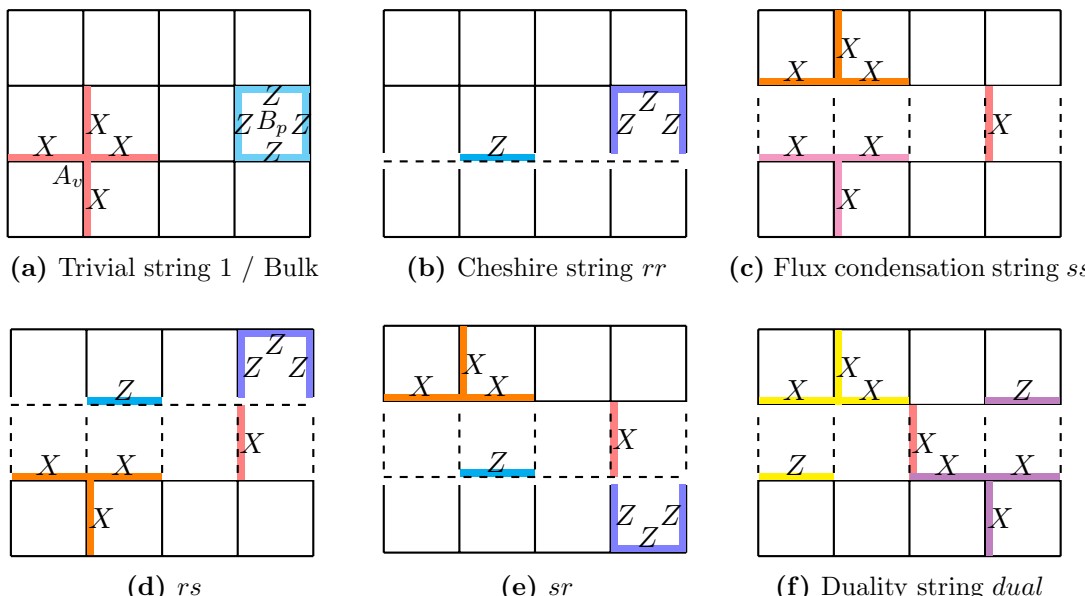

**(a)** Trivial string 1 / Bulk     **(b)** Cheshire string $rr$     **(c)** Flux condensation string $ss$

**(d)** $rs$        **(e)** $sr$        **(f)** Duality string $dual$

Figure 1: 1d LEEs of a 2D Toric Code.

Here, the first and the last 1d LEEs are invertible, in the sense that they can be fused back uniquely to the trivial string 1, such as $dual \times dual = 1$. The remaining four 1d LEEs are non-invertible and cannot be fused uniquely back to the identity. Remarkably, all six distinct types of 1d LEEs can be derived by using Cheshire string $rr$ and duality strings $dual$ as fusion building blocks. For instance, fusing $rr$ and $dual$ yields the $rs$ string, and fusing $dual$ and $rs$ results in the flux condensation string $ss$. Detailed structure the 1d finite depth circuit used to fuse these 1d LEEs, as well as the sequential quantum circuit to generate them, can be found in Ref. [19].

A more nontrivial fusion process involves two non-invertible 1d LEEs, where the fusion results are generally not unique. In this sense, it is similar to fusing two 0d non-abelian anyons in a non-abelian topological order. However, the distinction between fusing non-invertible 1d LEEs and 0d non-abelian anyons lies in the "coefficient", which is no longer a mere number but a decoupled 1D theory [3, 23], with possibly degenerate ground states in one-to-one correspondence with the fusion outcome.

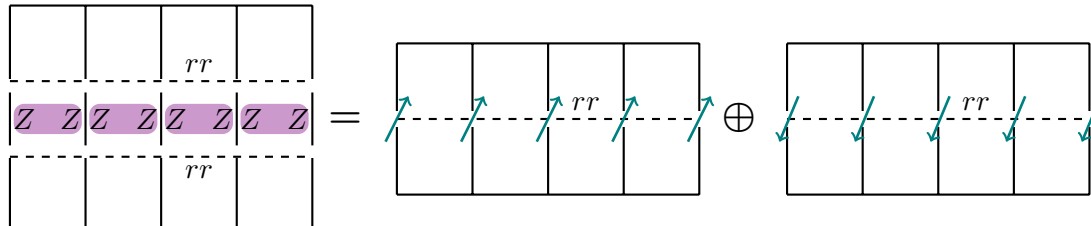

Figure 2: Fusion of two Cheshire strings $rr$.

For instance, consider fusing two Cheshire strings $rr$ within a 2D Toric Code, as shown in Fig. 2. Here, the vertex terms $A_v$ are removed along the two dashed $rr$ strings, and a polarization term $-Z$ is added on every dashed bond. As a result, the edges between the two $rr$ strings are decoupled from the rest of the system. The plaquette terms $B_p$ between the two Cheshire strings become ferromagnetic $ZZ$ couplings between neighboring edges, as shown by the purple blocks in Fig. 2. Therefore, there is a two-fold degeneracy of the fusion outcome, each labeled by the all-up or all-down ferromagnetic spins on the internal

edges. The corresponding fusion rule can be written as:

$$\overline{\quad\quad\quad\quad\quad\quad}rr \atop \overline{\quad\quad\quad\quad\quad\quad}rr \;\; = \; \text{-}\mathcal{Z}\text{-}\text{-}\mathcal{Z}\text{-}\text{-}\mathcal{Z}\text{-}\text{-}\mathcal{Z}\text{-}rr \;\oplus\; \text{-}\mathcal{Z}\text{-}\text{-}\mathcal{Z}\text{-}\text{-}\mathcal{Z}\text{-}\text{-}\mathcal{Z}\text{-}rr \;\;, \quad (2)$$

where the Cheshire strings $rr$ are shown by parallel solid lines, and the ferromagnetic states are explicitly shown as the "coefficient" of the fusion outcome. We can also write the fusion rule as $rr \otimes rr = (|\Uparrow\rangle \oplus |\Downarrow\rangle)rr$, where we use $|\Uparrow\rangle \equiv |00\cdots 0\rangle$ and $|\Downarrow\rangle \equiv |11\cdots 1\rangle$ to represent the ferromagnetic states. Later we will demonstrate the physical consequences of the "coefficient" being a 1d phase rather than a number: it can host its own domain wall excitations, which will couple to the 0d LEEs in the fusion outcome.

|  | 1 | $rr$ | $ss$ | $rs$ | $sr$ | $dual$ |
|---|---|---|---|---|---|---|
| 1 | 1 | $rr$ | $ss$ | $rs$ | $sr$ | $dual$ |
| $rr$ | $rr$ | $(|\Uparrow\rangle \oplus |\Downarrow\rangle)rr$ | $rs$ | $(|\Uparrow\rangle \oplus |\Downarrow\rangle)rs$ | $rr$ | $rs$ |
| $ss$ | $ss$ | $sr$ | $(|\Rightarrow\rangle \oplus |\Leftarrow\rangle)ss$ | $ss$ | $(|\Rightarrow\rangle \oplus |\Leftarrow\rangle)sr$ | $sr$ |
| $rs$ | $rs$ | $rr$ | $(|\Rightarrow\rangle \oplus |\Leftarrow\rangle)rs$ | $rs$ | $(|\Rightarrow\rangle \oplus |\Leftarrow\rangle)rr$ | $rr$ |
| $sr$ | $sr$ | $(|\Uparrow\rangle \oplus |\Downarrow\rangle)sr$ | $ss$ | $(|\Uparrow\rangle \oplus |\Downarrow\rangle)ss$ | $sr$ | $ss$ |
| $dual$ | $dual$ | $sr$ | $rs$ | $ss$ | $rr$ | 1 |

Table 1: Fusion rules of 1d LEEs of 2D Toric Code. Here $|\Rightarrow\rangle \equiv |++\cdots+\rangle$ and $|\Leftarrow\rangle \equiv |--\cdots-\rangle$ denote the ferromagnetic states with all the edges in the $+$ or $-$ states.

All other fusion rules can be obtained in a similar manner. Here the full set of fusion rules of 1d LEEs of a 2D Toric Code are listed in Table. 1, as originally shown in Ref. [18]. The difference is that here we explicitly denote the "coefficient" as ferromagnetic states rather than some numbers. We use $|\Rightarrow\rangle \equiv |++\cdots+\rangle$ and $|\Leftarrow\rangle \equiv |--\cdots-\rangle$ to represent the ferromagnetic states with all the internal edges in the $|+\rangle$ or $|-\rangle$ states, which appears when we try to fuse two all flux condensation string $ss$ together.

## 2.2  1d LEEs with domain walls or endpoints

The 1d LEEs can carry 0d point-like excitations along their length, such as domain walls between two 1d LEEs of the same type or boundaries between different 1d LEEs. To classify the 0d point like excitations along 1d LEEs, we can again utilize the unitary transformations. We consider two 0d point-like excitations to be equivalent if they can be connected through a local unitary transformation. Mathematically, these 0d point-like excitations are referred to as morphisms between the 1d LEEs. Combined with the 1d LEEs, they form a 2-category mathematical structure [18, 24, 25]. In this section, we explore the types of domain walls and boundaries between 1d LEEs of the 2D Toric Code and examine the fusion of these 0d excitations along the 1d LEEs using 0d unitary transformations.

### 2.2.1  Chershire string $rr$ with domain walls or endpoints

First, consider the domain walls of a Cheshire string $rr$, which is a morphism from the Cheshire string to itself. Evidently, the charge $e$ cannot generate any non-trivial 0d domain wall since a local unitary circuit can be found to condense $e$ onto the Cheshire string. As illustrated in Fig. 3 (a), a charge excitation $e$ trapped by a vertex $-A_v$ term near the Cheshire string $rr$ can be eliminated by a local unitary transformation $Z$, represented by the cyan diamond. On the other hand, the magnetic flux $m$ is confined on the Cheshire

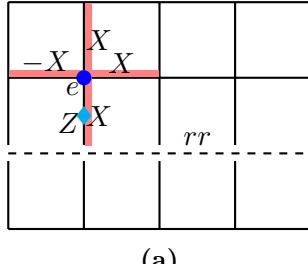
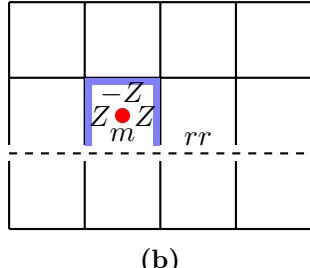

|  (a)  |  (b)  |

Figure 3: Possible domain walls of a Chehsire string $rr$, with (a) a charge $e$ or (b) a flux $m$ attached to it. The cyan diamond in (a) represents a local unitary transformation conjugated by a $Z$ operator on that edge.

string $rr$, implying it cannot be removed by any local transformation, as depicted in Fig. 3 (b).

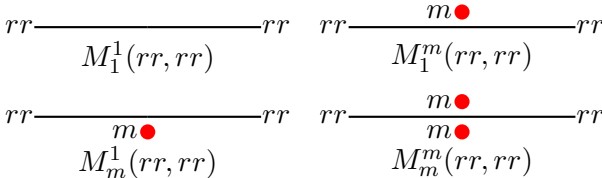

Figure 4: Domain walls of a Cheshire string $rr$.

Therefore, nontrivial excitations can be created by pulling the magnetic flux $m$ onto the Cheshire string $rr$ from either side of the string. The four kinds of morphisms from a Cheshire string $rr$ to itself are shown in Fig. 4. We denote these four kinds of morphisms as $M_1^1(rr,rr)$, $M_1^m(rr,rr)$, $M_m^1(rr,rr)$ and $M_m^m(rr,rr)$, where the $rr$ in the brackets means it is a morphisms from $rr$ to $rr$, and the superscript and subscript labels the anyons pulled to the Cheshire string.

The topological nature of these domain walls or morphisms can be seen from its nontrivial braiding with the non-confined particles on the 1d LEEs. In the present simple example, the magnetic flux $m$ confined on the Cheshire string has nontrivial braiding with the charge $e$ that is condensed. Therefore, we can create a condensed charge $e$ from the Cheshire string $rr$, let it wind around the given point and re-condense it on the string, as shown in Fig. 5. This is the so-called half-braiding process discussed in Refs. [26, 27]. A minus sign is obtained if there is a flux $m$, which is unchanged under any local unitary transformation around the flux $m$. The minus sign can be used to detect whether there is a flux morphism $m$ on either side of a Cheshire string.

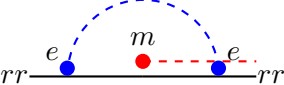

Figure 5: Detection of domain wall along a Cheshire string $rr$.

The anyon labels $\alpha$ and $\beta$ in $M_\beta^\alpha(rr,rr)$ are actually not unique for a given morphism. If $\alpha$ and $\alpha'$ are equivalent up to anyons that condense on the Cheshire string $rr$, they can be connected by a local unitary transformation and therefore label the same morphism. A simple example is $M_1^f(rr,r) = M_1^m(rr,rr)$, where $f$ is the fermion excitation in Toric Code:

$$rr\overset{m\bullet}{\rule{3cm}{0.4pt}}rr \;=\; rr\overset{f\bullet}{\rule{3cm}{0.4pt}}rr \quad . \tag{3}$$

Another interesting case is the Cheshire string $rr$ endpoints, which can be seen as a morphism from the trivial string 1 to the Cheshire string $rr$ or vice versa. It is easy to see that there are two different kinds of boundaries between the trivial and the Cheshire strings, as shown in Fig. 6. The two kinds of boundaries correspond to whether a flux $m$ is attached to the endpoint, and are denoted as $M_1^1(1, rr)$ and $M_1^m(1, rr)$, respectively.

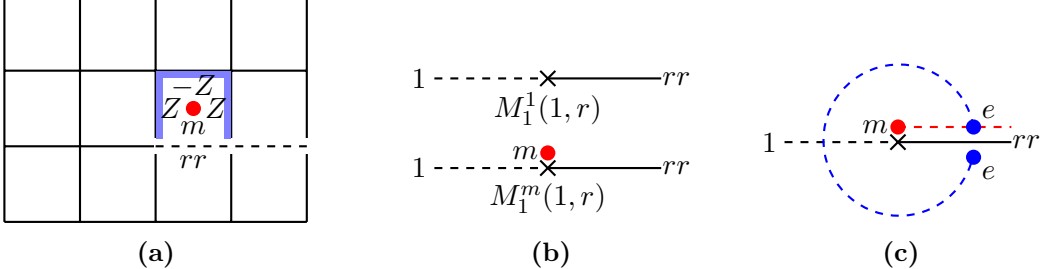

(a)                     (b)                     (c)

Figure 6: Endpoints between trivial string 1 and Cheshire string $rr$. (a) and (b) show the two possibilities by attaching a flux $m$ at the endpoint. (c) shows how to detect the endpoint morphisms through tunneling charge $e$.

Similar to the morphisms from Cheshire string $rr$ to itself, the topological nature of the two kinds of endpoints in Fig. 6 can be seen by its nontrivial braiding with the condensed $e$ particle. We can tunnel a condensed charge $e$ out of the Cheshire string, let it wind around the left endpoint, and retouch the bottom of the Cheshire string. We always get a minus sign if there is a magnetic flux $m$ at the endpoint, no matter what local unitary we apply around it.

### 2.2.2  $e - m$ duality string with endpoints

As stated in 2.1, all the 1d LEEs in 2D Toric Code can be obtained by fusing the Cheshire string $rr$ with the duality string $dual$. In a similar manner, to get all the domain walls and boundaries of 1d LEEs, we can first study the domain walls and boundaries of the Cheshire string $rr$ and the duality string $dual$, and then fuse them together. We already considered the domain walls of a Cheshire string $rr$ and now we turn to the duality string $dual$.

The number of distinct domain walls connecting a duality string to itself is identical to those for a trivial string, as the duality string is an invertible 1d LEE. They can be labeled by the anyons in the bulk $1, e, m, f$ on either the top side or bottom side of the string. The only difference is that a charge $e$ on top of the string will become a flux $m$ after going through the duality string, e.g. $M_1^e(dual, dual) = M_m^1(dual, dual)$.

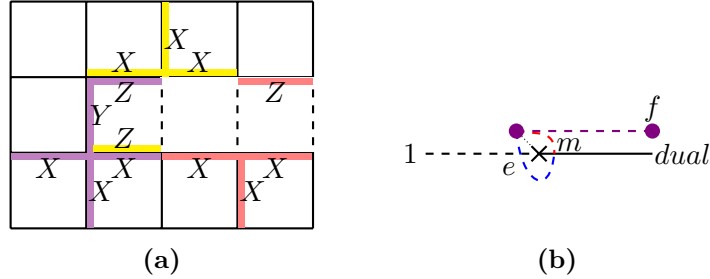

(a)                     (b)

Figure 7: Endpoint between a trivial string 1 and a duality string $dual$, with (a) showing its lattice realization and (b) illustrating the absorption of a fermion $f$ at the endpoint.

On the other hand, something nontrivial happens when the duality string has

endpoints. It is well know that there is a Majorana fermion at the endpoint of a duality string [18, 28, 29]. Fig. 7 (b) gives a simple picture of what happens at the endpoint of a duality string. A fermion $f$ can be annihilated or created at the boundary, by first creating a pair of charge $e$, moving one of the $e$ around the endpoint to become a flux $m$, and then fusing with the remaining charge $e$. In this sense, the fermion $f$ is condensed at the endpoint of a duality string.

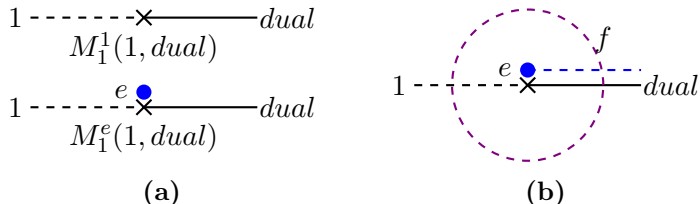

(a)             (b)

Figure 8: Boundaries between trivial string 1 and duality string $dual$, with (a) showing the two possibilities by attaching a charge $e$ at the endpoint, and (b) showing the detection of the endpoint morphisms through tunneling a fermion $f$ .

As a result, attaching a fermion $f$ at the endpoint does not change its type and there are only two kinds of morphisms between the trivial string and the duality string – those with or without a charge $e$ (or equivalently, a flux $m$) attached to it, as shown in Fig. 8 (a). The two kinds of endpoint morphisms can be distinguished by winding a fermion string around the endpoint, as also shown in Fig. 8 (b).

### 2.2.3   Other kinds of domain walls and boundaries

All other types of 1d LEEs, along with their domain walls or endpoints, can be obtained by fusing Cheshire and duality strings, as summarized in Table 2. We note that different types of morphisms between any two given 1d LEEs, $A$ and $B$, can be connected by attaching anyons from the bulk ($e$ and $m$ in the Toric Code) to the connection point between $A$ and $B$, either from the upper or lower side of the 1d LEEs. This idea is already demonstrated in earlier examples in Figs. 4, 6 and 8. Therefore, Table 2 lists only the generating operations of these morphisms, which involve pulling anyons from the bulk to the boundary from either the upper or lower side of the strings. By composing the anyon configurations listed in the table, one can generate the complete set of morphisms at the domain walls or boundaries between $A$ and B.

Here, we provide examples of how the results in Table 2 are derived. First, by fusing the duality string $dual$ from above onto the Cheshire string $rr$, we obtain the domain walls of the $sr$ 1d LEE:

$$
\begin{array}{c} dual \rule{2cm}{0.4pt} dual \\ rr \underset{m^b\,\circleddash}{\overset{m^a\,\circleddash}{\rule{2cm}{0.4pt}}} rr \end{array}
\quad = \quad
sr \underset{m^b\,\circleddash}{\overset{e^a\,\circleddash}{\rule{2cm}{0.4pt}}} sr \quad , \tag{4}
$$

where $a, b = 0$ or 1 represent the possibilities of presence or absence of an anyon, indicated by the shaded circles. Using the same idea, we derive the domain walls of $rs$ and $ss$. These morphisms can be detected using half braiding in a similar way as discussed for the domain walls on the Cheshire string $rr$.

To get the endpoints of the 1d LEEs, we can fuse an open Cheshire string $rr$ with an

| | 1 | $rr$ | $ss$ | $rs$ | $sr$ | $dual$ |
|---|---|---|---|---|---|---|
| $1$ | $e\,\bullet$ ---- <br> $m\bullet$ ---- | $m\bullet$ —✕— | $e\,\bullet$ —✕— | --✕-- | --✕-- | $e\,\bullet$ —✕— |
| $rr$ | — | $m\bullet$ —— <br> $m\bullet$ —— | —✕— | $m\bullet$ —✕— | —✕— $m\bullet$ | —✕— |
| $ss$ | — | — | $e\,\bullet$ —— <br> $e\,\bullet$ —— | —✕— $e\,\bullet$ | $e\,\bullet$ —✕— | —✕— |
| $rs$ | — | — | — | $m\bullet$ —— <br> $e\,\bullet$ —— | —✕— | $m\bullet$ —✕— |
| $sr$ | — | — | — | — | $e\,\bullet$ —— <br> $m\bullet$ —— | $e\,\bullet$ —✕— |
| $dual$ | — | — | — | — | — | $e\,\bullet$ —— <br> $m\bullet$ —— |

Table 2: Domain walls and boundaries (morphisms) between different kinds of 1d LEEs in 2D Toric Code. Only the generating operations of pulling anyons are shown here, as discussed in the main text.

open duality string $dual$, e.g.

$$
\begin{array}{c}
1\;\text{----}\;m^a\!\!\!\!\times\!\!\uparrow \;\text{——}\; rr \\
\qquad\quad \big( e^a \big) \\
1\;\text{------}\;\times\!\!\downarrow \;\text{——}\; dual
\end{array}
\;=\;\; 1\;\text{------}\;\times\!\!\text{——}\;rs \;. \tag{5}
$$

The $m$ anyon at the endpoint of the $rr$ string can wind around the endpoint of the $dual$ string and become an $e$ anyon which can condense on $rr$. Therefore, when fused with an open $dual$ string, the different types of endpoints of $rr$ become equivalent. Indeed, $rs$ has only one kind of endpoint morphisms as both charge $e$ and flux $m$ can condense on its boundary. As there is a Majorana fermion at the endpoint of the duality string, there will also be an extra degeneracy when fusing $rs$ string with endpoints, as shown latter.

To get the endpoint morphisms of a magnetic flux condensation string $ss$, we can use a closed duality loop to wrap a Cheshire string $rr$:

$$
1\;\text{---}\Big(\; m^a\!\!\!\!\times \;\begin{array}{c} \text{——}\,dual \\ \text{——}\;rr \\ \text{——}\,dual \end{array}
\;=\;\; 1\;\text{-----}\;e^a\!\!\!\!\times\;\text{——}\,ss \;. \tag{6}
$$

Finally, by fusing duality strings with endpoints and the Cheshire string $rr$ without endpoints, we get all other kinds of morphisms between different types of 1d LEEs. For example,

$$
\begin{array}{c}
1\;\text{------}\;\times\!\!\uparrow\;\text{——}\,dual \\
\qquad\quad e^a \\
rr\;\text{——}\;m^a\!\!\!\downarrow\;\text{——}\;rr \\
\qquad\quad m^b
\end{array}
\;=\;\; rr\;\text{——}\;m^b\!\!\!\!\times\;\text{——}\,sr \;. \tag{7}
$$

The magnetic flux on top of the Cheshire string disappears after the fusion, because it can first locally go through the duality defect to become a charge $e$, and then winds back to condense on the Cheshire string. As a result, there are only two kinds of morphisms between $rr$ and $sr$, which corresponds to whether a magnetic flux $m$ is attached from

below or not. Similarly,

$$
\begin{array}{c}
1 \dashleftarrow\!\!\times\!\!\rule[0.5ex]{2em}{0.4pt} dual \\[-0.3em]
rr\rule[0.5ex]{0.5em}{0.4pt}\dfrac{m^a \varobslash}{m^b \varobslash}\rule[0.5ex]{0.5em}{0.4pt} rr \quad = \quad rr\rule[0.5ex]{3em}{0.4pt}\!\!\times\!\!\rule[0.5ex]{2em}{0.4pt} ss \quad , \\[-0.3em]
1 \dashleftarrow\!\!\times\!\!\rule[0.5ex]{2em}{0.4pt} dual
\end{array}
\tag{8}
$$

i.e., there is only one kind of morphism between the Cheshire string $rr$ and its dual version $ss$.

## 2.3 Fusion of domain walls and endpoints along 1d LEEs

Just like anyons and 1d LEEs can fuse with each other, 0d domain walls and endpoints can also fuse along 1d LEEs. Similar to fusing two anyons in the bulk, fusing domain walls and boundaries can be achieved with local unitary transformations around the 0d excitations. Again we mainly focus on fusing morphisms along the Cheshire strings $rr$ and the duality strings $dual$. Other fusions can be obtained by fusing $rr$ and $dual$ together with their domain walls and endpoints.

Fusing the magnetic fluxes $m$ morphism along a Cheshire string is trivial, which is exactly the same as in the bulk where $m \times m = 1$.

$$
rr\rule[0.5ex]{0.5em}{0.4pt}\dfrac{m\bullet\dashrightarrow\bullet m}{rr}\rule[0.5ex]{0.5em}{0.4pt}rr \quad = \quad rr\rule[0.5ex]{4em}{0.4pt}rr \quad .
\tag{9}
$$

On the other hand, fusing the endpoints of Cheshire strings $rr$ is much more interesting. First, consider fusing the endpoint from Cheshire string $rr$ to trivial string 1 with the endpoint from trivial string 1 to Cheshire string $rr$, as shown in Fig. 9.

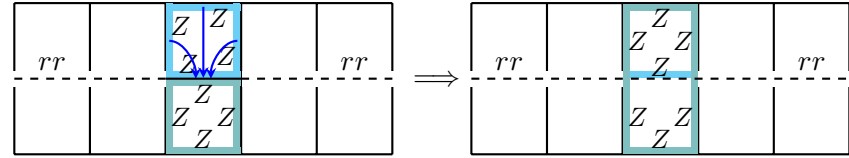

Figure 9: Local unitary transformation to fuse endpoints $M_1^1(rr, 1)$ and $M_1^1(1, rr)$. The blue arrows represent controlled-Not gates from the starting point to the endpoint of the arrows.

A local unitary circuit is used to fuse the two endpoints together into a domain wall from $rr$ to $rr$, as shown by the blue arrows in Fig. 9. Here, every blue arrow represents a controlled-Not gate $|0\rangle\langle 0|_c + |1\rangle\langle 1|_c X_t$, whose control qubit (denoted by $c$) is the starting point of the arrow and target qubit (denoted by $t$) is the endpoint of the arrow. After this local unitary transformation, the two $B_p$ stabilizers represented by the cyan and teal plaquettes on the left hand side of Fig. 9 are transformed to the stabilizers of the Cheshire string $rr$, as shown on the right hand side of Fig. 9. The cyan term becomes a single $Z$ stabilizer and therefore turns the trivial segment between the two Cheshire strings also into a charge $e$ condensate. However, the original teal plaquette term becomes the product of two plaquette terms, and therefore allowing a two-fold degeneracy: no flux at all or two fluxes attached to the upper and lower side of the fused domain walls. The fusion rule can be written as:

$$
rr\rule[0.5ex]{2em}{0.4pt}\!\!\times\!\!\dfrac{1}{\;}\!\!\times\!\!\rule[0.5ex]{2em}{0.4pt}rr \quad = \quad rr\rule[0.5ex]{4em}{0.4pt}rr \quad + \quad rr\rule[0.5ex]{0.5em}{0.4pt}\dfrac{m\bullet}{m\bullet}\rule[0.5ex]{0.5em}{0.4pt}rr \quad .
\tag{10}
$$

This fusion rule can also be easily verified as follows. Before the fusion, there is a two-fold degeneracy between the two open Cheshire strings $rr$, which can be distinguished by a string operator tunneling a charge $e$ from left Cheshire $rr$ to the right Cheshire $rr$. The value of the string operator can be either $+1$ or $-1$ depending on which ground state is chosen, and is independent of the local unitary transformation we used in Fig. 9 to fuse the endpoints. When the value of the string operator is $+1$, the fused domain wall should to trivial; and when it is $-1$, the fused domain wall should carry magnetic fluxes $m$ on both sides of the string to recover this $-1$ sign.

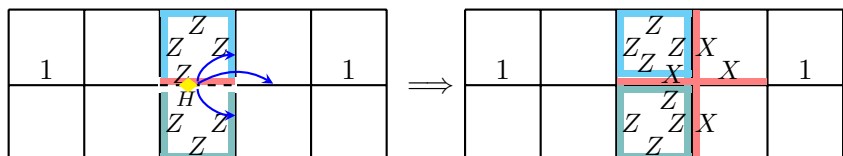

Figure 10: Local unitary transformation to fuse endpoints $M_1^1(1, rr)$ and $M_1^1(rr, 1)$. Here the blue arrows are still controlled-Not gates and the yellow diamond represents a local unitary conjugated by Hadamard gate $H$.

Another way to fuse the endpoints of the Cheshire strings is to reverse the order to consider $M_1^1(1, rr) \times M_1^1(rr, 1)$, as shown in Fig. 10. This time, to fuse the two trivial strings together, we need a two-step local unitary transformation. First we apply a Hadamard gate on the dashed bond as shown by the yellow diamond, which is followed by the controlled-Not gates indicated by the blue arrows. After this local unitary transformation, the red $Z$ term becomes the vertex term $A_{v'} = \prod X$ to the right of the original dashed bond, and the two truncated plaquette terms become full plaquette terms. However, the vertex term to the left of the dashed bond is still absent, which means the fusion result is a 0d LEE of an equal weight superposition of a charge and no charge, i.e.:

$$1 \text{----}\times\!\!\xrightarrow{rr}\!\!\times\text{----} 1 \;=\; 1 \text{--------------} 1 \;+\; 1 \text{--------}\overset{e\,\bullet}{\text{--------}} 1 \;. \qquad (11)$$

This result can be easily understood as the charge is condensed along the Cheshire segment before fusion. When the Cheshire segment is shrunk to a point, it gives rise to the degeneracy (not protected any more).

We call all of these morphisms, whose fusion outcomes are not unique, as non-invertible morphisms. For abelian topological orders, all of the domain wall morphisms from a 1d LEE to itself will be invertible. The non-invertible morphisms will only appear at the boundary between different 1d LEEs.

Another interesting example is the fusion of the endpoints of a duality string, which is also a non-invertible morphism as a result of the Majorana fermion at the endpoint. Fig. 11 shows an explicit circuit used to fuse the boundaries $M_1^1(dual, 1)$ and $M_1^1(1, dual)$ into the domain wall morphisms from $dual$ to itself. A three-step local unitary transformation is used here. The first step (blue) and the second step (cyan) of the circuit are again some controlled-Not gates similar to those in Fig. 9. The third gate (red diamond) is the phase gate $S = \begin{pmatrix} 1 & 0 \\ 0 & i \end{pmatrix}$ which transforms $X \to Y$, $Y \to -X$, $Z \to Z$.

After the action of the circuit, the yellow stabilizer becomes a single $Y$ stabilizer and the underlying edge is decoupled. On the other hand, the purple term becomes a complex new term, which is the product of two kinds of stabilizers of the standard duality string as shown in Fig. 1 (f) (yellow and purple). Physically, the anyon excitation corresponding to the yellow term in Fig. 1 (f) can be either a charge on the upper side of the string or a flux on the lower side of the string. Similarly, anyon excitations corresponding to the

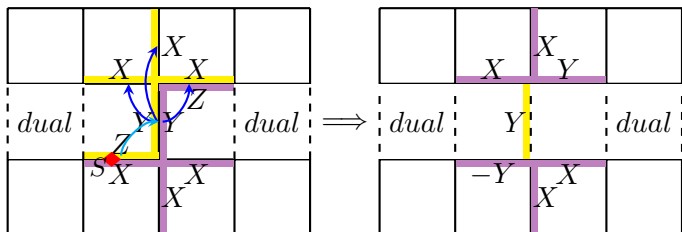

Figure 11: Circuit used to fusing the endpoints $M_1^1(dual, 1)$ and $M_1^1(1, dual)$, the circuits are acted in the order of blue-cyan-red. Here the blue and cyan arrows represent controlled-Not gates and the red diamond represents the phase gate $S$.

purple term in Fig. 1 (f) can be either a flux on the upper side of the string or a charge on the lower side of the string. As a result, a fermion excitation on either side of the string will not violate the new purple term in Fig. 11, which is a product of the above two terms. Therefore, the fusion rule reads as:

$$
\underset{dual}{\overset{1}{\times\text{--}\times}}_{dual} \;=\; \underset{dual}{\quad\quad}_{dual} \;+\; \underset{dual}{\overset{f\ \bullet}{\quad\quad}}_{dual} \;. \tag{12}
$$

By fusing a duality string either above or below the expression above, we can get the fusion rule of $M_1^1(1, dual) \times M_1^1(dual, 1)$:

$$
1\text{----}\underset{}{\overset{dual}{\times\quad\times}}\text{----}1 \;=\; \text{--------------}1 \;+\; \text{-----}\underset{}{\overset{f\ \bullet}{f}}\text{------}1 \;. \tag{13}
$$

Other horizontal morphisms fusion rules along 1d LEEs can be obtained by fusing the duality string $dual$ and the Cheshire string $rr$ together. For example, the fusion of a morphism from $rr$ to $ss$ and another morphism from $ss$ to $rr$ can be derived as:

$$
rr\underset{}{\overset{ss}{\quad\times\quad\times\quad}}rr \;=\; \frac{1\text{----}\overset{dual}{\times\quad\times}\text{----}1}{1\text{----}\underset{dual}{\times\quad\times}\text{----}1}rr
$$

$$
=\; \underset{}{\quad\quad}rr \;+\; \overset{m\bullet}{\quad\quad}rr \;+
$$

$$
\underset{m\bullet}{\quad\quad}rr \;+\; \overset{m\bullet}{\underset{m\bullet}{\quad\quad}}rr \;.
$$

(14)

Here we used the fact that a fermion $f$ is equivalent to a magnetic flux $m$ along a Cheshire string $rr$. The results illustrates that there is an extra degeneracy at the interface between rough and smooth boundary.

# 3 Fusing 1d LEEs with domain walls and endpoints

In the previous sections, we discussed the 1d LEEs (without 0d morphisms), the 0d morphisms between the 1d LEEs, as well as their respective fusion rules. However, we haven't yet addressed the fusion of two 1d LEEs with possibly nontrivial 0d morphisms on top. This is the topic of this section.

As was done in the previous sections, we use quantum circuit to give concrete meaning to this type of fusion. As discussed previously, the fusion of 1d LEEs is achieved with 1d finite depth circuits, and the fusion of 0d morphisms along a 1d LEE is achieved with 0d local unitaries. To fuse 1d LEEs with 0d morphisms on top, we apply the finite depth

circuit that fuses 1d LEEs without morphisms on either side of the morphism and can add extra 0d unitaries near the location of the morphism. But there is a complication. Since finite depth circuits can be used to create / modify domain walls along 1d LEEs, different choices of fusion circuits can result in different fusion outcome when the 1d LEEs carry domain walls / endpoints without affecting the fusion result of 1d LEEs without morphisms. As we will see below, this is the case not only for boundaries between different 1d LEEs where the fusion circuits on the two sides can be independently chosen, but for domain walls on the same 1d LEEs as well where the nontrivial coefficient in the fusion of non-invertible 1d LEEs can be used to control the fusion outcome of the morphism.

In this section, with the 2D Toric Code as an example, we discuss how 1d LEEs with 0d morphisms are fused with each other. In particular, we show that the 0d morphisms before fusion will become the morphisms of the fused 1d LEEs as well as domain walls of the coefficient. We also discuss how the different choices of finite depth 1d circuits for fusing 1d LEEs leads to different fusion result of the morphisms. Specifically, we discuss two kinds of 1d circuit for fusing Cheshire strings $rr$ with or without domain walls. We discuss the physical interpretation of the fusion result in these two cases and show that one of the 1d circuits can be easily generalized to 3D Toric Code.

### 3.1 Fusing Cheshire strings $rr$ with domain walls and endpoints

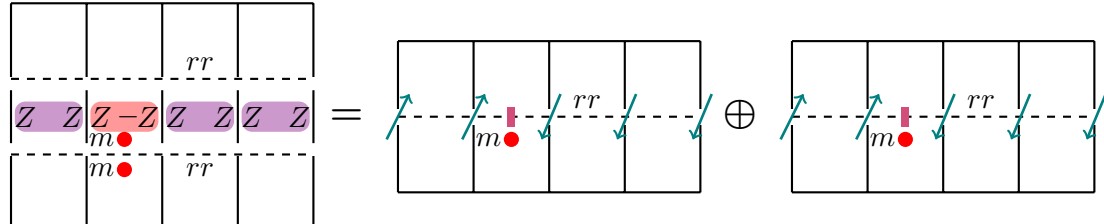

Figure 12: Fusing two Cheshire strings $rr$ with domain walls.

We first consider fusing two Cheshire strings $rr$ with domain walls along them, e.g. fusing a Cheshire $rr$ with domain wall $M_1^1(rr, rr)$ and another Cheshire $rr$ with domain wall $M_m^m(rr, rr)$ as shown in Eq. (15). We need to require that the fusion circuit in the bulk of the 1d LEEs be the same as that without any morphisms. According to 2.1, we are actually doing nothing in the bulk of the Cheshire strings $rr$ to fuse them together, as the edges between the two Cheshire strings $rr$ are already decoupled from the bulk. Therefore, the magnetic flux $m$ on top of the upper layer and below of the lower layer will remain invariant. On the other hand, a magnetic flux between the two Cheshire strings $rr$ will change a plaquette term from $ZZ$ to $-ZZ$, as illustrated by the red plaquette term in Fig. 2. As a result, it becomes a domain wall of the "coefficient" after fusion. The fuse rule can be written as:

$$
\begin{array}{l}
rr\text{———}rr \\
rr\underset{\overline{m\bullet}}{\overset{m\bullet}{\text{———}}}rr
\end{array}
= rr\underset{m\bullet}{\text{- - -}}rr \oplus rr\underset{m\bullet}{\text{- - -}}rr \ . \quad (15)
$$

Note that there is a magnetic domain wall in the "coefficient" of each of the fusion outcomes. This example shows that the 0d morphisms before the fusion not only affect the 0d morphisms of the fusion outcome, but also affect the domain walls of the "coefficient" 1d ferromagnetic phase. In this sense, the "coefficient" of the fusion rule in Eq. (2) is indeed not just a number, but the ground space of a 1d ferromagnetic chain. The "coefficient" and its domain walls can also have their own fusion rules, cf. Ref [1].

The next interesting example is to fuse Cheshire strings $rr$ with endpoints, as shown in Fig. 13 or Eq. (16). As stated in the last section, there are two different kinds of endpoint morphisms, $M_1^1(1, rr)$ and $M_m^1(1, rr)$. In Fig. 13, we consider fusing two open Cheshire strings without magnetic flux $M_1^1(1, rr)$. Different from fusing infinite long Cheshire string $rr$ as in Fig. 2, at the endpoints, the edges between two $rr$ are still coupled to the bulk through a plaquette term $B_p$.

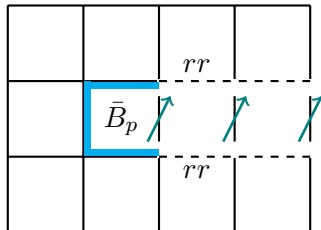

Figure 13: Fusing Cheshire strings with endpoints. $\bar{B}_p$ is the product of $Z$ operators of the cyan bonds.

To see the consequence of this remaining coupling, we can detect whether there is a morphism at the left endpoint, as we did in Fig. 6. We first generate a charge from the upper side of the upper Cheshire string $rr$, let it wind around the left endpoints and touch the bottom of the lower Cheshire string $rr$. This operator should map the ground state to itself up to a $\pm 1$ sign as the charge is condensed on the string.

The value of such a charge string operator is equal to the truncated plaquette term $\bar{B}_p = \pm 1$, as shown in Fig. 13. As a result, the two channels of the fused Cheshire strings carry different kinds of endpoint morphisms $M_1^1(rr, rr)$ and $M_1^m(rr, rr)$, as shown in Eq. (16). Moreover, since the un-truncated plaquette term takes eigenvalue $+1$, the morphism we get at the endpoint is entangled with the spins in the coefficient.

$$
\begin{array}{l}
1 \text{-------}\times\text{\rule{1cm}{0.4pt}}rr \\
1 \text{-------}\times\text{\rule{1cm}{0.4pt}}rr
\end{array}
= 1 \text{-------}\times\text{\textcircled{$\nearrow$}--\textcircled{$\nearrow$}-}rr \oplus 1 \text{-----}\overset{m}{\underset{\bullet}{\times}}\text{\textcircled{$\swarrow$}--\textcircled{$\swarrow$}-}rr \quad . \tag{16}
$$

When the spins in the coefficient are in the up state, $\bar{B}_p = 1$ and the fusion outcome is a Cheshire string $rr$ with trivial endpoint morphism $M_1^1(1, rr)$; when the spins in the coefficient are in the down state, $\bar{B}_p = -1$ and the fusion outcome is a Cheshire charge $rr$ with nontrivial morphism $M_m^1(1, rr)$, which is attached with a flux $m$. Later we show that a different choice of the finite-depth fusion circuit fully decouples the coefficient and the morphism of the fusion result.

Finally, we can also fuse a shorter Cheshire string with a longer Cheshire string:

$$
\begin{array}{l}
1 \text{-------}\times\text{\rule{1cm}{0.4pt}}rr \\
rr\text{\rule{1.5cm}{0.4pt}}rr
\end{array}
= rr\text{\rule{1cm}{0.4pt}}\text{-\textcircled{$\nearrow$}--\textcircled{$\nearrow$}-}rr \oplus rr\text{\rule{1cm}{0.4pt}}\overset{m}{\bullet}\text{-\textcircled{$\swarrow$}--\textcircled{$\swarrow$}-}rr \quad . \tag{17}
$$

## 3.2  Effect of different choices of fusion circuit on 0d morphisms

Choosing different 1d finite-depth circuits to fuse the 1d LEEs can result in different fusion rules when they carry nontrivial 0d morphisms. As an example, we again consider the problem of fusing two Cheshire strings $rr$. The circuit used in Fig. 2 and Fig. 12 is a trivial circuit. We can add a finite depth circuit that tunnels an $m$ anyon along the length of the Cheshire string. In this simple case, the new circuit does not change the fusion result of two Cheshire strings with no domain walls or the fusion result of two Cheshire strings

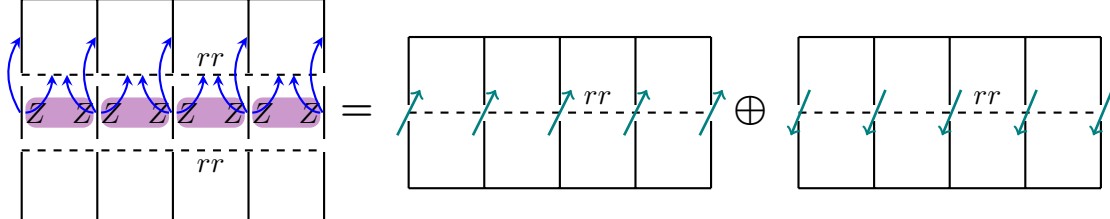

Figure 14: Fusing two Cheshire strings $rr$ with a modified circuit. The blue arrows represent controlled-Not gates.

with domain walls. It will change the fusion result of Cheshire strings with endpoints by adding an $m$ anyon to each endpoint.

A more interesting case is where the change in finite depth circuit changes the fusion rule even when the fusion is between two Cheshire strings with only domain walls but no endpoints. Readers referring to Ref. [19] may find that the circuit used there to fuse two Cheshire strings $rr$ together is different from the one used in Fig. 2 and Fig. 12. In Fig. 2 and Fig. 12, no non-trivial unitary action was needed to fuse the two strings together. In Ref. [19], a finite depth circuit controlled by the internal legs is used, as shown in Fig. 14. Here, the blue arrows in Eqs. (18) and Fig. 14 indicate controlled-Not gates controlled by the internal legs $|0\rangle\langle 0|_c + |1\rangle\langle 1|_c X_t$. All the blue arrows commute with each other and this is a finite-depth circuit. Physically, this circuit tunnels a magnetic flux $m$ along the length of the string on top of the upper Cheshire string, when the internal legs are in the down state. This is therefore a controlled-tunneling circuit, controlled by the ferromagnetic state of the coefficient. (The circuit in Fig. 14 is slightly different from that in Ref. [19] – the circuit in Ref. [19] has one controlled-Not gate per unit cell while the circuit in Fig. 14 has three – but their actions are equivalent.)

When there are no domain walls along the Cheshire strings before fusion, the fusion rule is exactly the same as in Eq. (2):

$$
\begin{array}{c}
\text{(diagram)}\,rr \\
\underline{\qquad\qquad}\,rr
\end{array}
\;=\;
\text{(diagram)}\,rr \;\oplus\; \text{(diagram)}\,rr \quad . \tag{18}
$$

On the other hand, the fusion rules of Cheshire strings $rr$ with 0d domain walls as shown in Eqs. (15) will change now under the different circuit defined in Fig. 14:

$$
rr\,\text{(diagram)}\,rr \;=\; rr\,\text{(diagram)}\,rr \;\oplus\; rr\,\text{(diagram)}\,rr \quad . \tag{19}
$$

The circuit will create a magnetic flux on top of the upper Cheshire string whenever there is a magnetic domain wall on the coefficient. With this circuit, the magnetic flux between two Cheshire strings $rr$ in Eq. (19) will not only affect the domain walls of the ferromagnetic "coefficient", but also change the morphisms above the upper Cheshire string.

Similarly, the fusion result of the endpoint morphisms in Eq. (16) becomes

$$
\begin{array}{c}
1\text{-----}\times\,\text{(diagram)}\,rr \\
1\text{-----}\times\underline{\qquad}\,rr
\end{array}
\;=\;
1\text{-----}\times\,\text{(diagram)}\,rr \;\oplus\; 1\text{-----}\times\,\text{(diagram)}\,rr \quad , \tag{20}
$$

where the endpoint morphisms are always trivial regardless of the "coefficient".

To consistently understand both fusion results, we note that the circuit defined in Figs. 2 and 14 are different up to a finite depth circuit which creates a flux $m$ above the

fused Cheshire string when there is a domain wall of the "coefficient". It changes the fusion result as:

$$rr \underset{m^b}{\overset{m^a}{\underline{\hspace{3cm}}}} rr \;\Rightarrow\; rr \underset{m^b}{\overset{m^{a+\rho-\sigma}}{\underline{\hspace{3cm}}}} rr \quad, \tag{21}$$

and

$$1 \underset{\phantom{m^a}}{\overset{m^a}{\text{-------}\times\underline{\hspace{2cm}}}} rr \;\Rightarrow\; 1 \underset{\phantom{m^{a+\sigma}}}{\overset{m^{a+\sigma}}{\text{-------}\times\underline{\hspace{2cm}}}} rr \quad, \tag{22}$$

where $\sigma, \rho = 0$ or $1$ represent the spins on the "coefficient" ferromagnetic state. This change in the fusion rule of 1d LEEs is consistent with the fusion rule of 0d LEEs along the 1d LEEs, as illustrated by the commutativity of the following diagrams,

$$
\begin{array}{ccc}
rr \underset{m^b \quad rr \quad m^d}{\overset{m^a \qquad\qquad m^c}{\underline{\hspace{4cm}}}} rr & = & rr \underset{m^{b+d}}{\overset{m^{a+c}}{\underline{\hspace{3cm}}}} rr \\[2mm]
\Downarrow & & \Downarrow \\[2mm]
rr \underset{m^b \quad rr \quad m^d}{\overset{m^{a+\rho-\sigma} \qquad m^{c+\delta-\rho}}{\underline{\hspace{4cm}}}} rr & = & rr \underset{m^{b+d}}{\overset{m^{a+c+\delta-\sigma}}{\underline{\hspace{3cm}}}} rr \quad,
\end{array}
\tag{23}
$$

and

$$
\begin{array}{ccccc}
rr \underset{1}{\overset{m^a \qquad\qquad m^b}{\underline{\times\text{------}\times}}} rr & = & rr \overset{m^{a+b}}{\underline{\hspace{2.5cm}}} rr & + & rr \underset{m}{\overset{m^{a+b+1}}{\underline{\hspace{2.5cm}}}} rr \\[2mm]
\Downarrow & & \Downarrow & & \Downarrow \\[2mm]
rr \underset{1}{\overset{m^{a-\sigma} \qquad m^{b+\rho}}{\underline{\times\text{------}\times}}} rr & = & rr \overset{m^{a+b+\rho-\sigma}}{\underline{\hspace{2.5cm}}} rr & + & rr \underset{m}{\overset{m^{a+b+1+\rho-\sigma}}{\underline{\hspace{2.5cm}}}} rr \quad.
\end{array}
\tag{24}
$$

We also note that the 0d domain walls on the "coefficient" are invariant regardless of the 1d fusion circuits.

Which 1d circuit should we choose to fuse two Cheshire strings $rr$ together? We note that the two different choices of 1d circuits here have different merits in generalizing to more general problems. Here we denote the circuit used in Fig. 2 as *Circ1*, and the circuit used in Ref. [19] and Fig. 14 as *Circ2*. The difference between the two circuits can be summarized as follows.

1. The fusion results of *Circ1* are symmetric under horizontal reflection, whereas the fusion results of *Circ2* typically are not. This difference is evident when fusing two Cheshire strings $rr$, with a flux $m$ between them. In the case of *Circ1*, this flux $m$ becomes a domain wall of the "coefficient" but does not affect the domain walls of the fused strings. In contrast, under *Circ2*, the flux $m$ between the two $rr$ strings is tunneled to the upper side of the fused string, breaking the horizontal reflection symmetry of the original configuration.

2. On the other hand, *Circ2* has the advantage that the "coefficient" is completely decoupled from the bulk and fused 1d LEE after fusion, even when the Cheshire strings $rr$ have endpoints, as shown in Eq. (20). This allows us to safely disregard the "coefficient" ferromagnetic state after fusion. In contrast, under the action of *Circ1*, the endpoint morphisms of the fused 1d LEE remain entangled with the "coefficient", as shown in Eq. 16.

3. Due to disentanglement of the "coefficient", *Circ2* can be easily generalized to higher dimensions like the 3D Toric Code model, which is discussed in detail in the next section 4.

4. The fusion rule under *Circ2* can be ungauged to recover the fusion rule of the symmetry breaking chains, which was derived in Ref. [1]. There, the fusion rule of two symmetry breaking GHZ states (denoted as SB) can be written as

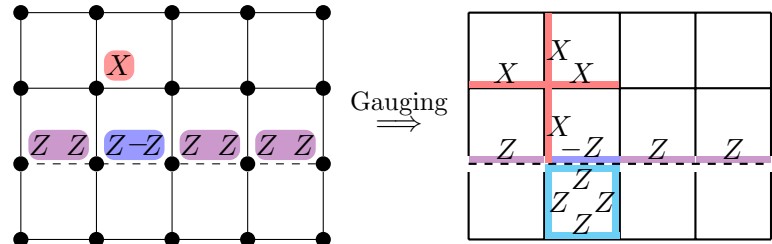

$$(25)$$

where the blue arrows again represent controlled-Not gates. After the fusion circuit, the upper SB state becomes a $\mathbb{Z}_2$ "coefficient" while the lower SB state remains invariant. And when there is a domain wall before fusion on the lower SB state (denoted by the red dot), it becomes a domain wall of both the $\mathbb{Z}_2$ "coefficient"and the fused SB string.

Figure 15: Gauging of symmetry breaking phase with domain walls. Corresponding operators before and after the gauge mapping are marked in the same color.

The fusion rule in Eq. (25) can be mapped to the fusion rule in Eq. (19) through a simple gauge mapping. To see this, consider that gauging a 1d SB defect on a symmetric bulk state results in the Cheshire string $rr$ in a Toric Code [19]. Similarly, a domain wall along an SB state corresponds to a $-Z$ term along the Cheshire string $rr$, as shown in Fig. 15. If we conjugate the the $-Z$ term with $X$, the two plaquette terms $B_p$ on the two sides of the $-Z$ terms will be transformed to $-B_p$. It becomes clear that a $-Z$ term corresponds to a process of attaching two fluxes $m$ on both upper and lower side, i.e. $M_m^m(rr, rr)$.

$$\text{SB} \underset{\bullet}{\rule{2cm}{0.4pt}} \text{SB} \implies rr \underset{\bullet m}{\overset{\bullet m}{\rule{2cm}{0.4pt}}} rr \ . \qquad (26)$$

By applying this gauge mapping, it is straightforward to see that Eq. (25) maps directly to Eq. (19). This outcome is expected, given that *Circ2* is independent of dimension.

5. However, the disentangling feature of *Circ2* cannot be naively generalized to more general 2D string-net models. For example, to decouple the endpoint morphisms from the "coefficient", which corresponds to attaching a non-abelian anyon on the 1d LEEs, a sequential quantum circuit is generally needed. Therefore, *Circ1* seems easier to be generalized to 2D non-abelian topological orders.

## 3.3 Fusing duality strings with domain walls and endpoints

We now consider fusing duality strings with domain walls and boundaries. Unlike the Cheshire strings $rr$, the duality string is an invertible 1d LEE, and fusing it with any

other 1d LEEs gives a single output channel. However, due to the noninvertible nature of the boundary of a duality string, there can be extra degeneracy caused by morphisms.

Consider the fusion of two open duality strings with endpoints, as shown in Fig. 16. Unlike the fusion of two closed duality strings, there is a two-fold degeneracy in the fusion result due to the Majorina fermions at the endpoints. A direct way to see the two-fold degeneracy is to construct two anti-commuting ribbon operators as shown in Fig. 16. $F_1$ creates a pair of fermions $f$ and then condenses them at the endpoint of one of the open duality strings. It anti-commutes with the $F_2$ operator that creates a pair of charge $e$, winds one of them around the endpoints and fuses the pair back to vacuum.

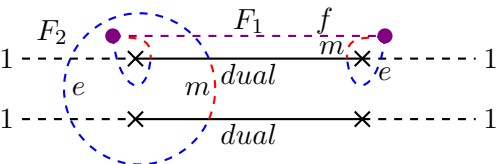

Figure 16: Two ribbon operators that commute with LEE stabilizers but anti-commute with each other.

As a result, the fusion of two duality strings with endpoints is an equal weight superposition of a trivial string 1 with no morphism or a trivial string 1 with a fermion:

$$
\begin{array}{l} 1\text{ -- -- -- } \times\!\!\!\text{-----}dual \\ 1\text{ -- -- -- } \times\!\!\!\text{-----}dual \end{array} = 1\text{ -- -- -- -- -- -- -- } 1 \oplus 1\text{ -- -- -- } \overset{f \,\bullet}{\text{-- -- -- -- }} 1 \ . \quad (27)
$$

The two-fold degeneracy can be distinguished by whether there is a fermion at the fused endpoints.

This degeneracy of fusion result can also be seen explicitly from the lattice model. To do this, we first write down the 1d circuit which can be used to fuse two duality strings, as derived in Ref. [19]. Fig. 17 (a) shows a set of stabilizers (and the translated copies of the terms shown) that stabilize a pair of duality strings, which can be seen by moving a charge $e$ or flux $m$ through the two strings. Here we choose a set of stabilizers such that the fusion result will not cause translation of the lattice [19, 30]. To fuse these two duality strings into a trivial string, we need a finite depth circuit as discussed in Ref. [19]. Here we slightly reformulate the 1d circuit such that the stabilizers after fusion become those of a standard Toric Code. The step-by-step 1d fusion circuit are given in Fig. 17 (a)-(d).

Now, to fuse duality strings with endpoints, we can use the above circuit in the bulk of the 1d LEEs, and only modify the 1d circuit locally around the endpoints. The new 1d circuit with an endpoint, as well as the transformation of stabilizers at the endpoint, are shown step-by-step in Fig. 18 (a)-(d). The red term in Fig. 18 (d) is the product of a vertex term and a plaquette term, and therefore allows either trivial anyon or a fermion $f$ as shown in Eq. (27).

Again, there is some freedom in choosing the 1d fusion circuit. For example, we can add a finite depth circuit to the circuit in Fig. 17, which tunnels a magnetic flux $m$ from the left endpoint to the right endpoint. After which the fusion rule of the endpoint morphisms becomes:

$$
\begin{array}{l} 1\text{ -- -- -- } \times\!\!\!\text{-----}dual \\ 1\text{ -- -- -- } \times\!\!\!\text{-----}dual \end{array} = 1\text{ -- -- } \overset{m\,\bullet}{\text{-- -- -- -- }} 1 \oplus 1\text{ -- -- } \overset{e\,\bullet}{\text{-- -- -- -- }} 1 \ . \quad (28)
$$

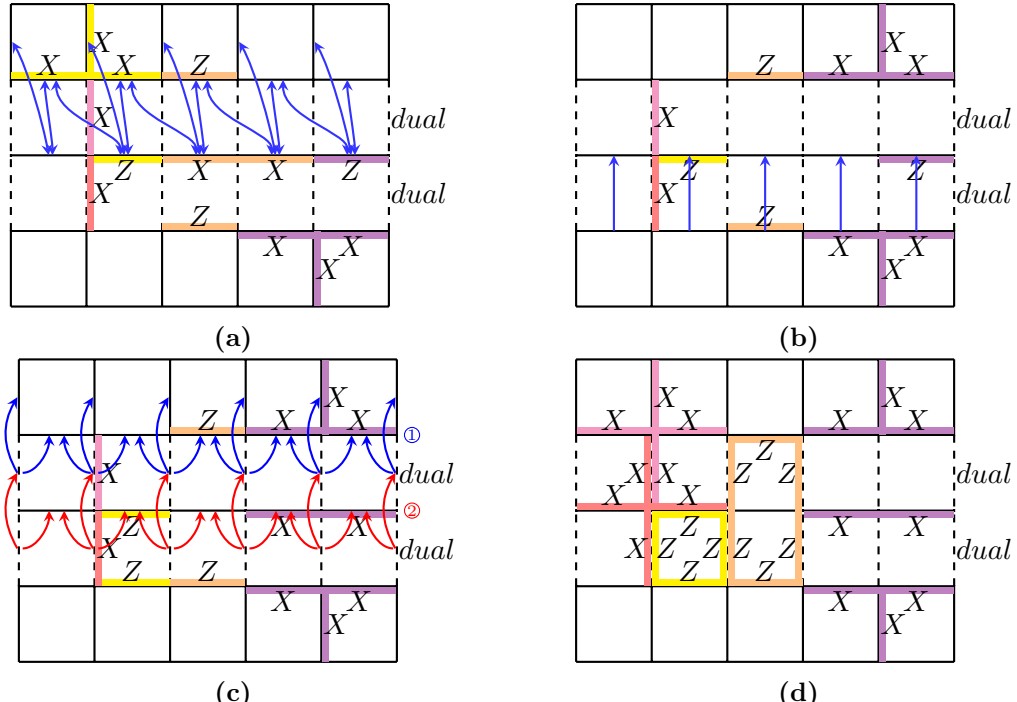

Figure 17: Lattice model of a pair of duality strings, with each stabilizer shown in the same color. A 1d circuit used to fuse the two duality strings together as well as the change of the stabilizers are shown step-by-step in the Figure. Here, the single-pointed-arrows again represent the controlled-Not gate $CX = |0\rangle\langle0|_c + |1\rangle\langle1|_c X_t$, while the double-pointed-arrows are controlled gates with the controlled-Not qubit conjagated by the Hadamard gate $XCX = H_c CX H_c$. The operations in (c) act in the order of blue-red, and from (b) to (c), we drop the $Z$ term in the purple stabilizer as there is already a $Z$ stabilizer represented by the yellow bond. The stabilizers in (d) can be reorganized into those for the standard Toric Code.

## 3.4 Fusing other 1d LEEs with domain walls or boundaries.

With the fusion rules of the Cheshire strings $rr$ and the duality strings $dual$, it is in principle easy to get all of the other fusion rules of 1d LEEs with domain walls or endpoints. Instead of deriving the full set of fusion rules, here we show some examples with interesting features. The fusion results are shown only for a specific fusion circuit.

For instance, fusing the duality string $dual$ with endpoint and the $rs$ string with endpoint results in the $ss$ string with multiple morphisms at the endpoint. This is because, similar to the endpoint of a duality string, there is also a degeneracy associated with the endpoint of the $rs$ string, which can be seen from the ribbon operators shown in Fig. 19. We therefore obtain a similar fusion rule as fusing two duality strings with endpoints in Eq. (27):

$$
\begin{array}{l}
1\text{-----}\times\text{------}dual \\
\hspace{3.5cm} = \quad 1\text{------}\times\text{------}ss \ \oplus \ 1\text{-----}\times^{e\,\bullet}\text{------}ss \quad . \quad (29) \\
1\text{-----}\times\text{------}rs
\end{array}
$$

This is another example where the degeneracy of the fusion outcome is caused by the morphisms – fusing closed duality and $rs$ strings simply gives $ss$.

More interestingly, when fusing open $rs$ and $sr$ strings together, there will be

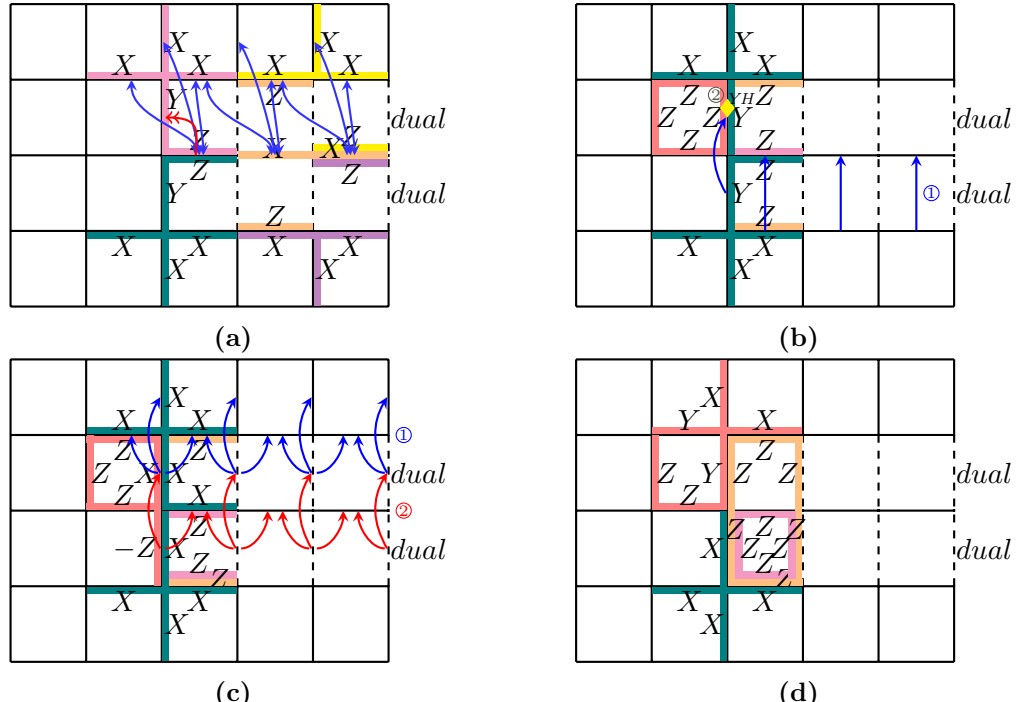

Figure 18: Lattice model of a pair of opened duality strings, with each stabilizer shown in the same color. A 1d circuit used to fuse the two opened duality strings as well as the change of the stabilizers are shown step-by-step in the Figure. Here, the red two-arrow in (a) is control gates with the X-control gate gate $XCY = H_c CY H_c$, where $CY = |0\rangle\langle 0|_c + |1\rangle\langle 1|_c Y_t$. The yellow diamond in (b) is $HY$. The operations in (c) act in the order of blue-red, and from (b) to (c), we drop the $Z$ term in the teal stabilizer as there is already a $Z$ stabilizer represented by the pink bond.

degeneracy caused by both the "coefficient" and the endpoint morphisms:

$$
\begin{aligned}
&
\begin{matrix}
1\text{-----}\times\text{------}sr \\
1\text{-----}\times\text{------}rs
\end{matrix}
\\[2pt]
={}&
\begin{matrix}
1\text{----}\times\text{------}dual \\
1\text{----}\times\!\!\oslash\text{--}\oslash\text{--}rr \\
1\text{----}\times\text{------}dual
\end{matrix}
\;+\;
\begin{matrix}
1\text{----}\times\text{------}dual \\
1\text{----}\times\!\!\oslash\text{--}\oslash\text{--}rr \\
1\text{----}\,m\!\bullet\text{------}dual
\end{matrix}
\\[6pt]
={}&
1\text{----}\left(\times\!\!\oslash\text{--}\oslash\text{--}rr\ dual\ dual\right)
\;+\;
1\text{----}\left(f\!\bullet\ \times\!\!\oslash\text{--}\oslash\text{--}rr\ dual\ dual\right)
\;+
\\[6pt]
&
1\text{----}\left(m\!\bullet\ \times\!\!\oslash\text{--}\oslash\text{--}rr\ dual\ dual\right)
\;+\;
1\text{----}\left(e\!\bullet\ \times\!\!\oslash\text{--}\oslash\text{--}rr\ dual\ dual\right)
\\[6pt]
={}&
1\text{------}\times\!\!\oslash\text{--}\oslash\text{--}ss
\;+\;
1\text{-----}\times\!\!\oslash\text{--}\oslash\text{--}ss\ (e\bullet)
\;+
\\[6pt]
&
1\text{-----}\times\!\!\oslash\text{--}\oslash\text{--}ss\ (e\bullet)
\;+\;
1\text{------}\times\!\!\oslash\text{--}\oslash\text{--}ss\ .
\end{aligned}
\tag{30}
$$

In this example, the fusion outcome is four-fold degenerate: two of them are caused by the "coefficient" and another two are caused by the morphisms.

And there are also cases where the "coefficient" is truly a number when fusing two $sr$

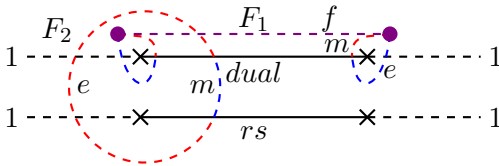

Figure 19: Two ribbon operators that commute with LEE stabilizers but anti-commute with each other.

strings with endpoints:

$$
\begin{array}{c}
1\text{-----}\times\!\!\!-\!\!\!-\!\!\!-\!sr \\
1\text{-----}\times\!\!\!-\!\!\!-\!\!\!-\!sr
\end{array}
= 1\text{-------}\times\!\!\!-\!\!\!-\!\!\!-\!sr \; + \; 1\text{-----}\!f\!\text{-}\!\times\!\!\!-\!\!\!-\!\!\!-\!sr
\tag{31}
$$

$$
= 2 \times \; 1\text{-------}\times\!\!\!-\!\!\!-\!\!\!-\!sr \;\; ,
$$

where we use the knowledge that a fermion $f$ at the endpoint of a $sr$ string is equivalent to trivial, as it can locally split to a charge $e$ and flux $m$ to condense on $sr$.

## 4  Discussion

The idea of fusing 1d LEEs as well as their 0d domain walls in 2D Toric code can be generalized to 1d LEEs and morphisms in 3Dtopological orders. In this section, we generalize the result by considering a 3D Toric Code model. Ref. [24] started the study of the 2-category structure of defects in 3D Toric Code. Working out all of the 1d LEEs and their fusion rules in 3D Toric Code is beyond the scope of this work. Here we mostly focus on the fusion of the Cheshire string, including those with domain walls and endpoints.

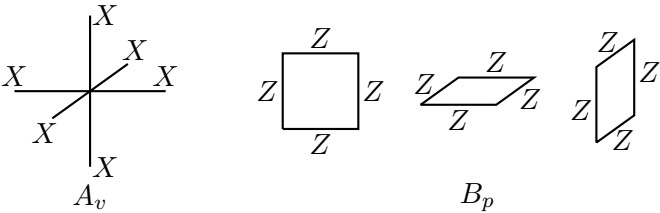

Figure 20: Lattice model of the 3D Toric Code model.

We start with the lattice realization of 3D Toric Code. The $\mathbb{Z}_2$ degrees of freedom are defined on the edges of the cubic lattice, and the commuting Hamiltonian terms are defined in Fig. 20. Different from 2D Toric Code, although the charge excitations $e$ which violate the vertex terms $A_v$ are still point-like excitations, the flux excitations $m$ which violate the plaquette terms $B_p$ are now loop-like excitations. To put the charge and flux excitations on the same ground, a 2-category structure of defects in 3D Toric Code was introduced [24]. The fundamental objects in the 2-category are the magnetic flux loop $m$ and the 1d Cheshire string (denoted as $r$ here). According to our definitions above, both the Cheshire string $r$ and the magnetic flux string $m$ are 1d LEEs that are stable under 1d finite depth quantum circuits. Here we focus on the fusion of 1d Cheshire strings together with their morphisms in the 3D Toric Code.

To fuse infinitely long Cheshire strings without any morphisms, we need a 1d circuit to decouple the edges connecting the two Cheshire strings from the rest of the system.

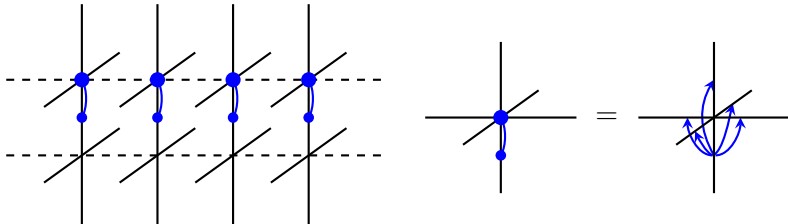

Figure 21: Fusing two Cheshire strings $r$ in 3D Toric Code.

This time, we can no longer use the circuit in Fig. 2, as the edges between the Cheshire strings are coupled to the bulk through the third dimension. Instead, we need to use a generalized version of the circuit defined in Fig. 14. The fusion rule is exactly the same as in 2D Toric Code:

$$r \; \begin{array}{c} \bullet \; \bullet \; \bullet \; \bullet \\ r \; \underline{\hspace{2cm}} \; r \end{array} \; r \;\; = \;\; \text{-}\oslash\text{-}\oslash\text{-}\oslash\text{-}\oslash\text{-} \; r \;\; \oplus \;\; \text{-}\oslash\text{-}\oslash\text{-}\oslash\text{-}\oslash\text{-} \; r \;\; . \qquad (32)$$

The difference comes in when considering morphisms. In 3D Toric Code, there are only two kinds of domain walls from a Cheshire string $r$ to itself, and only a single kind of endpoint between trivial string 1 and Cheshire string $r$. The nontrivial morphism between $r$ and itself is generated by winding a magnetic flux line around it, as shown in Fig. 22, which becomes trivial at the endpoint of a Cheshire string $r$.

$$r \; \underline{\hspace{3cm}} \; r \quad r \; \underline{\hspace{1cm}}^{m}_{\phantom{.}\mathsf{O}} \underline{\hspace{1.5cm}} \; r \quad 1 \; \text{-}\text{-}\text{-}\text{-}\text{-}\text{-}\text{-}\!\times\!\underline{\hspace{2cm}} \; r$$

Figure 22: Domain walls and endpoints of a Cheshire string $r$ in 3D Toric Code.

Making use of the same procedure as in 2D, the fusion rule of 1d Cheshire strings with morphisms can be derived as:

$$r \; \begin{array}{c} \bullet \; \bullet \; \mathsf{O} \; \bullet \; \bullet \\ r \; \underline{\hspace{2cm}} \; r \end{array} \; r \;\; = \;\; r \; \text{-}\oslash\text{-}\oslash\text{-}\!\!\rule[-0.4ex]{0.6pt}{1.6ex}\!\!\text{-}\oslash\text{-}\oslash\text{-} \; r \;\; \oplus \;\; r \; \text{-}\oslash\text{-}\oslash\text{-}\!\!\rule[-0.4ex]{0.6pt}{1.6ex}\!\!\text{-}\oslash\text{-}\oslash\text{-} \; r \;\; , \qquad (33)$$

$$r \; \begin{array}{c} \bullet \; \bullet \; \bullet \; \bullet \\ r \; \underline{\hspace{1cm}}_{\mathsf{O}}\underline{\hspace{1cm}} \; r \end{array} \; r \;\; = \;\; r \; \text{-}\oslash\text{-}\oslash\text{-}\!\!\rule[-0.4ex]{0.6pt}{1.6ex}_{\mathsf{O}}\!\!\text{-}\oslash\text{-}\oslash\text{-} \; r \;\; \oplus \;\; r \; \text{-}\oslash\text{-}\oslash\text{-}\!\!\rule[-0.4ex]{0.6pt}{1.6ex}_{\mathsf{O}}\!\!\text{-}\oslash\text{-}\oslash\text{-} \; r \;\; . \qquad (34)$$

Similar to fusing Cheshire strings $rr$ with magnetic flux $m$ in 2D Toric Code as in Eq. (19), there is an asymmetry of the fusion result coming from the 1d fusion circuit we choose. When there is a magnetic flux loop around the upper Cheshire string $r$, it will become a domain wall of the "coefficient" after the fusion. When there is a flux loop morphism around the lower Cheshire string $r$, it will affect both the "coefficient" domain walls and the morphism of the fusion result. Fusing Cheshire strings $r$ with endpoints has no difference from fusing infinite long strings:

$$1 \text{-}\text{-}\text{-}\text{-}\text{-}\text{-}\!\times\!\begin{array}{c} \bullet \; \bullet \\ \end{array}\! r \atop 1 \text{-}\text{-}\text{-}\text{-}\text{-}\text{-}\!\times\!\underline{\hspace{1cm}} r \;\; = \;\; 1 \text{-}\text{-}\text{-}\text{-}\text{-}\!\times\!\text{-}\oslash\text{-}\oslash\text{-} \; r \;\; \oplus \;\; 1 \text{-}\text{-}\text{-}\text{-}\text{-}\!\times\!\text{-}\oslash\text{-}\oslash\text{-} \; r \;\; . \qquad (35)$$

Interestingly, the fusion rules Eqs. (33),(34) and (35) are again consistent with the fusion rule of symmetry breaking phases in Eq. (25) up to a gauging procedure, as already mentioned in Sec. 3. To see this, we can do a similar gauge mapping of the lattice model

as we did in Fig. 15. The only difference is that the domain wall on the SB state will no longer become a pair of fluxes around the Cheshire, but now a ring of flux around the Cheshire, i.e.

$$\text{SB} \rule{2cm}{0.4pt}\!\!\bullet\!\!\rule{2cm}{0.4pt} \text{SB} \implies r \rule{2cm}{0.4pt}\!\!\overset{m}{\circ}\!\!\rule{2cm}{0.4pt} r \ . \tag{36}$$

One can check accordind to this gauge mapping rule, the fusion rule in Eq. (25) again maps to the fusion rule in Eq. (34). This is not surprising as fusing two SB strings together should be independent of dimension, and so are its gauged versions.

## 5 Conclusion

In this paper, we study the fusion of low-entanglement excitations in 2D Toric Code using quantum circuits. This includes the fusion of 1d LEEs, the fusion of 0d morphisms along a 1d LEE and the fusion of 1d LEEs with nontrivial morphisms on top.

We observe some interesting features in the fusion of 1d LEEs with nontrivial morphisms. First, since the 1d circuit used to fuse 1d LEEs without morphisms are not unique, when we use different circuits to fuse 1d LEEs with nontrivial morphisms we can get different results. Secondly, the fusion result of 1d LEEs with nontrivial morphisms may not be commutative between the two 1d LEEs. We find this to be true not only in 2D Toric Code but in 3D Toric Code as well. This is a bit surprising since in 3D, 1d LEEs can braid around each other, so we might expect their fusion to be symmetric under their exchange. This is the case for the fusion of anyons in 2D or higher dimensional topological phases. Anyons can braid around each other in 2 and higher dimensions and naturally their fusion is commutative. This is however no longer true for 1d LEEs. This feature already shows up in 1d LEEs in the trivial phase when we try to fuse 1d symmetry breaking chains with domain walls on top [1]. Moreover, we see that the non-commutative fusions in the trivial phase can be mapped to that in the Toric Code through the gauging map. Therefore, if we try to define a braided-fusion 2-category structure for the 1d LEEs in 3D or higher, we need to be more careful. For anyons, their fusion rule is automatically consistent with braiding. For 1d LEEs, to ensure the fusion is consistent with braiding, more structures must be built in.

## Acknowledgement

We are grateful for the inspiring discussions with Tian Lan and Linqian Wu. X.C. is supported by the Walter Burke Institute for Theoretical Physics at Caltech, the Simons Investigator Award (award ID 828078), the Institute for Quantum Information and Matter at Caltech, and the Simons Collaboration on "Ultra-Quantum Matter" (grant number 651438).

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
