# Peer review of "Fusion of Low-Entanglement Excitations in 2D Toric Code"

_SciPost Physics_

## Round 1 · Referee Report · Anonymous (Referee 2) · 2024-12-10

Strengths
-clear presentation
-detailed, explicit analysis
Report
This paper provides a systematic analysis of low entanglement excitations in topological order, in particular in 2+1d and 3+1d Z2 toric code. The work gives a clear framework for understanding such objects in lattice models.
1) As mentioned at the very beginning of the paper, excitations are usually used to refer to higher energy eigenstates of the same Hamiltonian. In fixed point models, these excitations are well-defined, but usually in a lattice model we only specify the ground state and the higher energy states can be very messy. Are your results only for fixed point models, or do you expect them to generalize to other gapped (non-commuting) Hamiltonians? Quasiparticles (O(1) energy) can be quasiadiabatically continued, but I'm not sure about these excitations with extensive energy.
2) "Given their low entanglement, the LEEs are potentially ‘condensable’ such that their condensation can still have low entanglement and potentially realize a different phase" If we view LEE as obtained by condensing anyons along a submanifold, is this different from condensing everywhere rather than along the submanifold? Although this story doesn't seem to work if you have different condensations from different sides i.e. rs, sr LEEs. Can you explain this point in more detail?
3) Does Fig 12 have an error? Should it be up/down on the left and down/up arrows on the right?
4) Above eq 15, do you mean Fig. 12 rather than Fig 2?
5) "First, since the 1d circuit used to fuse 1d LEEs without morphisms are not unique, when we use different circuits to fuse 1d LEEs with nontrivial morphisms we can get different results." Is there a canonical choice for the circuit? Can you comment on a general statement of what "different results" means? i.e. given two LEEs, is there an algebraic result about the distinct possible fusion rules?
Requested changes
Please respond to the comments/questions above and revise the manuscript as you feel appropriate.
Recommendation
Ask for minor revision
Strengths
Uncovers an interesting fusion rule of 1D Low-Entanglement Excitations and 0D morphisms in 2D and 3D toric code
Weaknesses
None
Report
The paper study the 0d and 1d Low-Entanglement Excitations in 2D and 3D toric codes. Using the language of quantum circuit, the authors discuss the fusion rule of low-entanglement excitations. In paticular, the non-commutative fusion rule in the 1d LEEs with nontrivial morphisms is very suprising and interesting. This implies a very detailed braided-fusion 2-category structure of Low-Entanglement Excitations in non-invertible topological phases.
The paper is clearly written, and contains new interesting results on algebra sturcture of 0d and 1d Low-Entanglement Excitations in 2D and 3D. Their fusion can be explicitly shown by constructing 0d unitary transformations or 1d circuits. It provides a insight into the algebra and entanglement nature of excitated states. Therefore, the referee recommends that the paper is published in SciPost.
Requested changes
None
Recommendation
Publish (easily meets expectations and criteria for this Journal; among top 50%)

---

## Editorial Decision

unknown